# A novel bioinformatics pipeline for the identification of immune inhibitory receptors as potential therapeutic targets

Akashdip Singh[1,2], Alberto Miranda Bedate[1,3†], Helen J von Richthofen[1,2†], Saskia V Vijver[1,2†], Michiel van der Vlist[1,2], Raphael Kuhn[4], Alexander Yermanos[1,4], Jürgen J Kuball[1,3], Can Kesmir[5], M Ines Pascoal Ramos[1,2], Linde Meyaard[1,2]*

[1]Center for Translational Immunology, University Medical Centre Utrecht, Utrecht University, Utrecht, Netherlands; [2]Oncode Institute, Utrecht, Netherlands; [3]Department of Hematology, University Medical Center Utrecht, Utrecht University, Urecht, Netherlands; [4]Department of Biosystems Science and Engineering, ETH Zurich, Zurich, Switzerland; [5]Theoretical Biology and Bioinformatics, Department of Biology, Utrecht University, Utrecht, Netherlands

*For correspondence:
L.Meyaard@umcutrecht.nl

†These authors contributed equally to this work

**Abstract** Despite major successes with inhibitory receptor blockade in cancer, the identification of novel inhibitory receptors as putative drug targets is needed due to lack of durable responses, therapy resistance, and side effects. Most inhibitory receptors signal via immunoreceptor tyrosine-based inhibitory motifs (ITIMs) and previous studies estimated that our genome contains over 1600 ITIM-bearing transmembrane proteins. However, testing and development of these candidates requires increased understanding of their expression patterns and likelihood to function as inhibitory receptor. Therefore, we designed a novel bioinformatics pipeline integrating machine learning-guided structural predictions and sequence-based likelihood models to identify putative inhibitory receptors. Using transcriptomics data of immune cells, we determined the expression of these novel inhibitory receptors, and classified them into previously proposed functional categories. Known and putative inhibitory receptors were expressed across different immune cell subsets with cell type-specific expression patterns. Furthermore, putative immune inhibitory receptors were differentially expressed in subsets of tumour infiltrating T cells. In conclusion, we present an inhibitory receptor pipeline that identifies 51 known and 390 novel human inhibitory receptors. This pipeline will support future drug target selection across diseases where therapeutic targeting of immune inhibitory receptors is warranted.

## eLife assessment

The authors presented a **valuable** bioinformatics pipeline for screening and identifying inhibitory receptors for potential drug targets. They provided **solid** evidence showing a sequential reduction in the search space through various screening tools and algorithms and demonstrated that this pipeline can be used to "rediscover" known targets. Further experimental validation on putative and unknown inhibitory receptors will strengthen the evidence reported in this work. This study will be of interest to bioinformaticians and computational biologists working on immune regulation, sequence screening, and target identification of immune checkpoint inhibitors.

## Introduction

In the past decade, checkpoint blockade therapy has revolutionised the treatment of cancer by releasing immune cells from inhibition in the tumour microenvironment. Approval of blocking antibodies targeting immune inhibitory receptors, i.e., cytotoxic T cell lymphocyte antigen 4 (CTLA-4)

(ipilimumab), the programmed cell death protein 1 (PD-1)/PD-L1 axis (nivolumab, atezolizumab), and lymphocyte-activation gene 3 (LAG-3) (*Chocarro et al., 2022*), resulted in significantly increased responses in previously hard-to-treat cancers, such as metastatic melanoma or non-small-cell lung cancer (*Andrews et al., 2019*). Despite these successes, most cancer patients do not respond to checkpoint therapy, or do not have a durable treatment response (*Sharma et al., 2017*). Furthermore, the central role for CTLA-4 and PD-1 in maintaining peripheral tolerance results in immune-related adverse events for many patients (*Wolchok et al., 2013*). Further understanding of tissue- or activation-specific expression of different inhibitory receptors can assist in limiting toxicities, in particular when receptors are also expressed on non-immune cells (*von Richthofen and Meyaard, 2023*). Additional benefit can be gained by targeting inhibitory receptors with more specialised functions or expression patterns, such as T cell immunoglobulin and mucin domain-containing receptor 3 (TIM-3) or T cell immunoreceptor with Ig and immunoreceptor tyrosine-based inhibitory motif (ITIM) domains (TIGIT) (*Schnell et al., 2020*).

We previously proposed that inhibitory receptors can be classified into distinct functional categories based on the regulation of their expression (*Rumpret et al., 2020*). Using mathematical modelling, we defined four categories: (1) Negative feedback receptor expression is induced by cellular activation to ensure a timely resolution of the response and limit excessive responses. (2) Threshold receptors are expressed on immune cells in the resting state and control the threshold for immune cell activation to provide context and prevent unnecessary or futile responses. (3) Threshold-disinhibition receptors are threshold receptors of which the expression is downregulated after activation. This allows for a more potent response after initial activation. (4) Threshold-negative feedback receptors are threshold receptors of which expression is further induced upon cellular activation.

It is reasonable to assume that therapeutic interference with inhibitory receptors of different functional categories would have different outcomes. In cancer, blocking threshold receptors will lower the activation threshold of immune cells with antitumour activities. Meanwhile, interference with negative feedback receptors will prolong the antitumour and cytolytic activities of immune cells.

The vast majority of currently characterised inhibitory receptors recruit phosphatases, such as SH2 domain-containing phosphatase (SHP-1), SHP-2, or SHIP-1 to their intracellular immunoreceptor via tyrosine-based inhibitory and/or switch motifs (ITIM/ITSMs) (*Long, 1999*). This allows these receptors to dephosphorylate and inactivate other signalling molecules, such as those recruited to the T cell receptor complex, although some receptors are also capable of relaying activating signals through an ITSM (*Dietrich et al., 2001*; *Ostrakhovitch and Li, 2006*). ITIM sequences consist of the consensus amino acid sequence (V|L|I|S)xYxx(I|L|V), where x can be any amino acid, in the intracellular domain of a protein. This domain can be extended to (V|L|I|S|T)xYxx(I|L|V) to also include ITSM sequences, as there are inhibitory receptors, such as PD-1, that rely partly on the ITSM for their inhibitory function (*Chemnitz et al., 2004*). So far, around 50 ITIM/ITSM containing inhibitory receptors have been described, although it has previously been estimated that over 1600 ITIM-containing molecules can be found in the human genome (*Daëron et al., 2008*).

More specific selection of potential drug targets out of this large collection of predicted receptors is required for further development and subsequent targeting in disease. Here, we set out to predict potential targets by developing a computational framework to identify putative inhibitory receptors and integrate three-dimensional structure predictions of proteins to those with a high likelihood of encoding functional inhibitory receptors. Furthermore, as inhibitory receptor expression follows specific dynamics and can play different roles in orchestrating immune responses (*Rumpret et al., 2020*), we analysed changes in the expression of putative inhibitory receptors in different immune cells and activation states. Our analysis will aid in the selection of inhibitory receptors as therapeutic targets for specific diseases, to minimise toxicity and maximise efficacy.

## Results

### Putative ITIM/ITSM-bearing immune inhibitory receptors can be found in the human genome

To identify putative novel inhibitory receptors, we first retrieved the protein sequences of all 96,457 protein-coding transcripts, corresponding to 19,353 individual human genes, annotated in the Ensembl database (release 105, December 2021) (*Cunningham et al., 2022*). We retrieved all deposited

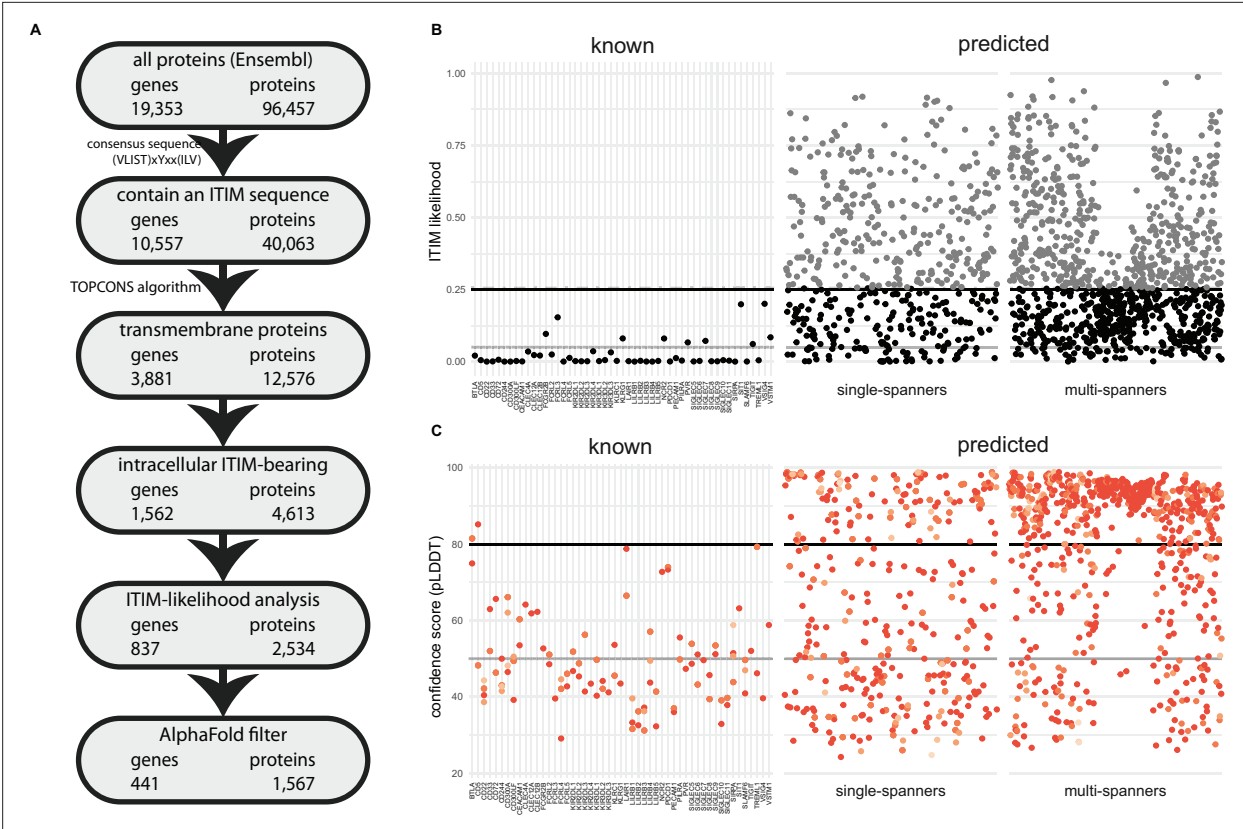

**Figure 1.** A novel bioinformatics approach for the identification of predicted inhibitory receptors. (**A**) Schematic overview of the bioinformatics pipeline with the number of unique genes and corresponding proteins remaining at every step. All amino acid sequences corresponding to a protein-coding transcript were retrieved from Ensembl. (**B**) Intracellular domains of identified proteins were permutated 10,000 times, and the number of immunoreceptor tyrosine-based inhibitory motif (ITIM) or immunoreceptor tyrosine-based switch motif (ITSM) occurrences were compared to the number of ITIMs in the original sequence to determine the likelihood of a specific intracellular domain containing an ITIM or ITSM. Threshold was determined based on known inhibitory receptors, and set at 0.25+ε, with ε being a random number between 0 and 0.01 to better predict the borderline predictions. The black line indicates 0.25 likelihood, the grey dotted line indicates 0.25+ε and the solid grey line indicates 0.05 likelihood. (**C**) Three-dimensional structure for all proteins was predicted using AlphaFold, and the average model prediction score (pLDDT) was determined for each individual ITIM or ITSM in the protein. Proteins with all ITIMs above 80 pLDDT were excluded. The black line indicates 80 pLDDT threshold and the solid grey line indicates 50 pLDDT. For **B** and **C**, one protein is plotted for every unique gene symbol for clarity.

The online version of this article includes the following figure supplement(s) for figure 1:

**Figure supplement 1.** A novel bioinformatics approach for the identification of predicted inhibitory receptors.

isoforms, as certain proteins might only contain an ITIM in a specific isoform. We then screened the entire protein sequence for the presence of an ITIM or ITSM, based on the consensus sequence (V|L|I|S|T)xYxx(I|L|V), resulting in 40,063 protein sequences from the initial 96,457. We next used the TOPCONS server, which is based on a consensus of several prediction tools (*Tsirigos et al., 2015*), to determine the membrane topology of all proteins with an ITIM/ITSM motif. TOPCONS predicted 12,576 out of the 40,056 input proteins to be integral membrane proteins, of which 7445 contained multiple transmembrane domains and the remaining 5131 are single-pass membrane proteins.

We next filtered proteins for the presence of the ITIM or ITSM sequence in an intracellular domain of the protein based on the predicted topology. This yielded with 4613 protein sequences bearing one or more intracellular ITIMs or ITSMs, which correspond to 1562 genes (*Figure 1A*, *Supplementary file 1*). To assess how well our pipeline reflects previously documented inhibitory receptors, we confirmed that all the 52 proteins known for their ITIM-mediated immune inhibitory effects, such as PD-1 or leukocyte-associated immunoglobulin-like receptor 1 (LAIR-1) (*Table 1*; *Rumpret et al., 2020*), were recalled successfully, with the exception of *MPIG6B*, encoding for the inhibitory receptor G6B on platelets and megakaryocytes (*de Vet et al., 2001*), due to an incorrect prediction of its transmembrane domain.

**Table 1.** Genes encoding previously described immunoreceptor tyrosine-based inhibitory motif (ITIM)-bearing inhibitory receptors.

| | | | | |
|---|---|---|---|---|
| BTLA | CLEC4A | KIR3DL2 | NCR2 | SIGLEC11 |
| CD22 | FCGR2B | KIR3DL3 | PDCD1 | SIRPA |
| CD244 | FCRL2 | KLRC1 | PECAM1 | SIT1 |
| CD300A | FCRL3 | KLRG1 | PILRA | SLAMF6 |
| CD300LF | FCRL4 | LAIR1 | PVR | TIGIT |
| CD33 | FCRL5 | LILRB1 | SIGLEC5 | TREML1 |
| CD5 | KIR2DL1 | LILRB2 | SIGLEC6 | VSIG4 |
| CD72 | KIR2DL2 | LILRB3 | SIGLEC7 | VSTM1 |
| CEACAM1 | KIR2DL3 | LILRB4 | SIGLEC8 | |
| CLEC12A | KIR2DL4 | LILRB5 | SIGLEC9 | |
| CLEC12B | KIR3DL1 | *MPIG6B* | SIGLEC10 | |

The composition of an ITIM/ITSM sequence is highly variable, as only three out of six amino acids positions are limited to specific amino acids. As a result of this, large proteins or proteins containing many tyrosines and hydrophobic residues are more likely to contain an ITIM/ITSM by chance (*Figure 1—figure supplement 1A*). We therefore determined the likelihood of encountering an ITIM/ITSM by chance in the intracellular region of the putative targets. To this end, we randomly permutated the amino acid sequences of the intracellular domains by shuffling their order using the Fisher-Yates algorithm. We did this 10,000 times for each receptor and assessed how often the permutated sequence contained more or an equal amount of ITIM/ITSMs compared to the original sequence. For 41 out of 51 known receptors, these odds were low (<5%). However, for 10 out of 51, we observed ITIM/ITSM sequences in the permutated sequence between 5% and ~25% of the time. Based on these findings, we decided to exclude novel targets when they were above this threshold of 25% (*Figure 1B*). Due to the nature of permutation testing, there is some variation in the individual likelihood values for each protein sequence. However, as they were generally below 0.25 in any given iteration, we decided to define this value as a threshold for inclusion. After this statistical filtering, we were left with 837 inhibitory receptor genes. As expected, the average length of the intracellular domains of the excluded proteins was much higher than that of the remaining targets (*Figure 1—figure supplement 1B*).

## AlphaFold structure predictions can assist in identifying likely functional ITIM/ITSMs

Protein interaction motifs are most commonly found in intrinsically disordered regions within proteins (*Tompa et al., 2014*), as structured regions are less accessible to binding with interacting partners. We therefore hypothesised that ITIM/ITSM sequences locate in disordered regions to allow for interaction with phosphatases and kinases. We therefore leveraged AlphaFold to infer three-dimensional structure predictions of the remaining inhibitory receptor candidates to determine where the ITIM/ITSM sequences were located in three-dimensional space (*Jumper et al., 2021*).

For 832 of the 837 remaining inhibitory receptor genes, we were able to retrieve the predicted structures using AlphaFold. For every residue in the three-dimensional model, AlphaFold defines a confidence score (pLDDT) between 0 and 100 pLDDT. Low scores indicate reduced confidence, and regions with a low score are more likely to be intrinsically disordered (*Tunyasuvunakool et al., 2021*). We determined the average pLDDT of all six residues encompassing the ITIM/ITSM sequence, what resulted in an overall confidence score for each intracellular ITIM/ITSM (*Figure 1C*) in all targets.

We found that 99 out of 101 ITIM/ITSMs of the 51 known receptors had low confidence scores, i.e., less than 80 pLDDT, with an average confidence score of 49.3 pLDDT. 64 of the ITIMs even had pLDDT values below 50, which means they are particularly likely to be intrinsically disordered (*Tunyasuvunakool et al., 2021*; *Figure 1—figure supplement 1C*). This result confirmed the hypothesis that ITIM/ITSMs are likely to be located in disordered regions in the protein and indicates that the AlphaFold

confidence score can be a valuable asset to determine the potential functionality of an ITIM sequence in predicted targets.

Based on the known receptors, we determined a threshold of 80 pLDDT and were left with 390 out of 832 putative ITIM-bearing inhibitory receptor genes likely to contain a functional ITIM/ITSM. Among these genes, 54.6% consisted of multi-spanning proteins, while the remaining 177 single transmembrane domain targets consisted of 145 (37.2%) type I and 32 (8.2%) type II receptors. In some cases, a single gene was predicted to have both single-spanning and multi-spanning isoforms. Most putative inhibitory receptors (275) were based on the canonical transcript, as annotated by Ensembl, while 115 were identified as a non-canonical isoform of the gene.

## Putative inhibitory receptors are expressed across immune cell subsets

To better characterise these receptors, we studied the expression patterns of the retrieved known and putative inhibitory receptor genes in various immune cells. We collected publicly available bulk RNA sequencing data of various immune cell subsets, both at resting state and after in vitro stimulation (*West et al., 2016*; *Calderon et al., 2019*; *Table 2*). We included all putative inhibitory receptors, including those that were identified in a non-canonical transcript of the gene. For this analysis, we considered a receptor as expressed when the expression was above the median overall gene expression in the sample, in either resting or activated state. From our set of inhibitory receptor genes, 2/51 known and 41/390 predicted genes were not detected in these datasets.

Almost all functionally characterised immune inhibitory receptors are type I transmembrane proteins, while only six are type II single spanners. So far, no multi-spanning receptors have been described to have an inhibitory function in the immune system. Also, many genes encoding multi-spanning proteins only showed very limited expression across immune cell subsets in the datasets we used for our analysis. Therefore, we analysed the multi-spanning targets (*Figure 2—figure supplement 1*) separately from the single-spanning novel targets (*Figure 2*).

We investigated the expression of 215 single-spanning receptor genes in these immune cell subsets and found that neutrophils (137 genes) and monocytes (120 genes) expressed the highest number of inhibitory receptor genes. Lymphoid cells expressed fewer inhibitory receptor genes compared to myeloid cells, with B and NK cells expressing 111 genes, followed by CD8[+] (106 genes) and CD4[+] (99 genes) T cells (*Figure 2B*). The relative contribution of known and novel targets in each subset varied between different immune cells. We found more novel targets being expressed in CD4[+] (86/99 genes, 86.9%) and CD8[+] T cells (88/106 genes, 84.3%), compared to neutrophils (98/137 genes, 71.5%). Neutrophils and monocytes uniquely expressed nine and three inhibitory receptor genes respectively, but the vast majority were shared between two or more immune cell subsets (*Figure 2C*).

Compared to the single spanners, we found slightly fewer multi-spanning inhibitory receptor genes expressed in the different immune cell subsets. Out of 197 predicted genes, neutrophils expressed the most multi-spanning inhibitory receptor genes (118 genes), while the other subsets expressed a similar number: Monocytes expressed 100 inhibitory receptor genes, followed by B cells (95 genes), CD4[+] (90 genes) and CD8[+] (90 genes) T cells, and NK cells (89 genes) (*Figure 2—figure supplement 1A and B*). Neutrophils (14 genes) expressed the most unique set of multi-spanning inhibitory receptors, while most other genes were shared between all subsets (*Figure 2—figure supplement 1C*).

**Table 2.** Number of samples for different resting and activated immune cell subsets.

| Cell type | Subsets | Resting | Activated | Stimulation | Duration |
|---|---|---|---|---|---|
| Neutrophils | | 3 | 10 | TSLP/*Staphylococcus aureus* | 4 hr/24 hr |
| Monocytes | | 4 | 8 | LPS | 24 hr |
| NK cells | | 6 | 6 | IL-2 | 24 hr |
| B cells | Naïve, memory, plasmablast | 13 | 10 | Anti-IgG/M+IL-4 | 24 hr |
| CD4 T cells | Naïve, effector, memory, regulatory, Tfh | 37 | 40 | Dynabeads +IL-2 | 24 hr |
| CD8 T cells | Naïve, effector, memory | 15 | 16 | Dynabeads +IL-2 | 24 hr |

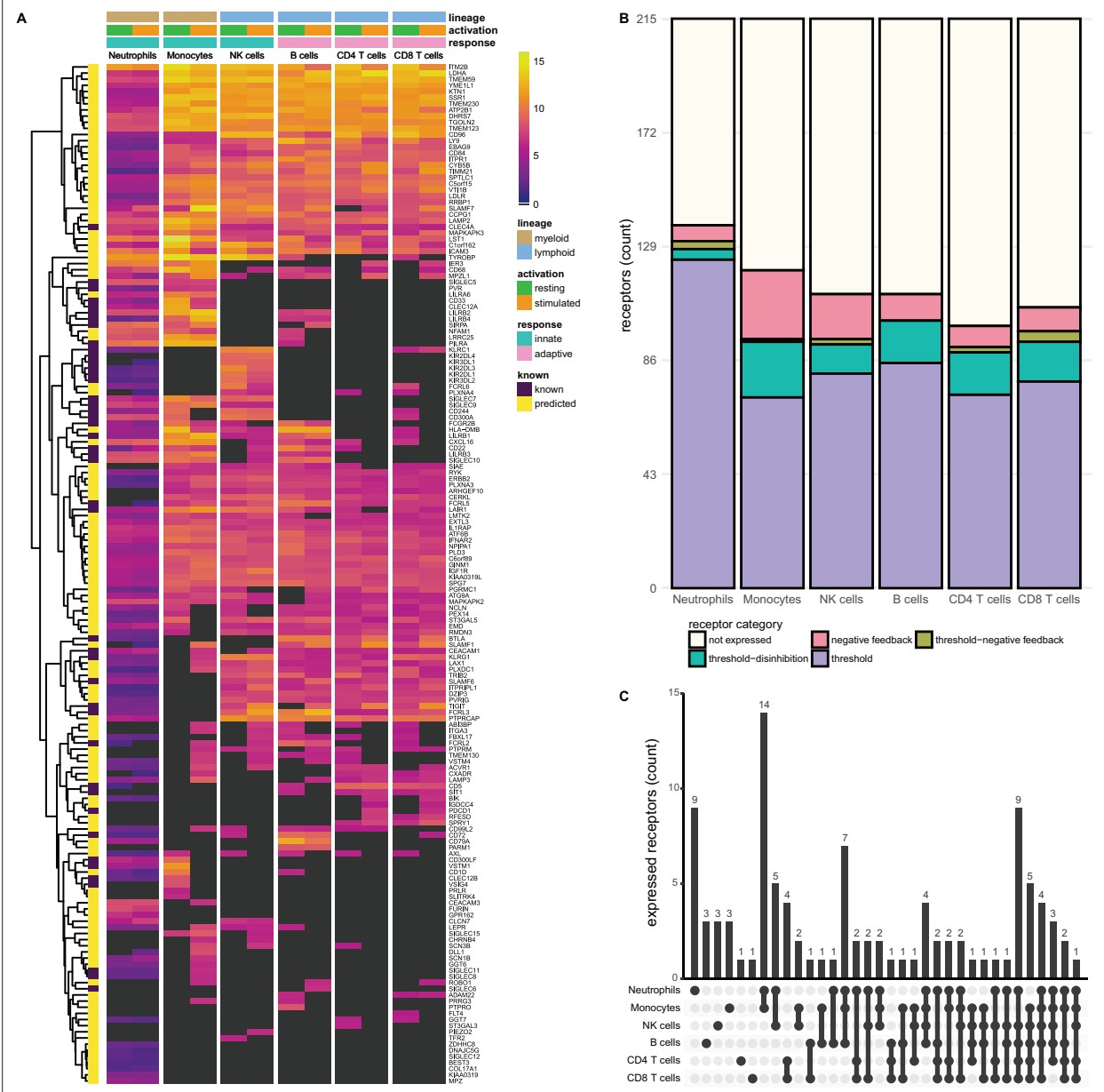

**Figure 2.** Known and predicted single-spanning inhibitory receptors are expressed in different cell types in the resting state and after activation. (**A**) Heatmap with normalised expression data for known and predicted single-spanning inhibitory receptor genes in different cell types, in the resting state or after activation. Receptors were considered not expressed (black) when expression was below median overall gene expression in the sample. Data for neutrophils was retrieved from a different source than the other cell types. (**B**) Novel and known receptors were classified into different functional categories based on changes, or lack thereof, in expression after stimulation. Threshold receptors were expressed at resting state, and did not change after activation (i.e. change in expression <0.5 log2 fold change). Threshold-negative feedback and threshold-disinhibition receptors were defined by >0.5 log2 fold change up- or downregulation, respectively, in expression after activation. Negative feedback receptors were absent in the resting state, but were expressed after activation. (**C**) Upset plot showing the number of single-spanning receptors that are expressed uniquely by individual immune cell subsets, or shared between subsets as indicated by connected circles. Sixty-four genes are expressed in all cell types (not depicted).

The online version of this article includes the following figure supplement(s) for figure 2:

**Figure supplement 1.** Known and predicted multi-spanning inhibitory receptors are expressed in different cell types in the resting state and after activation.

## Known and novel immune inhibitory receptors of different functional categories can be identified in immune cells

We next assigned the putative inhibitory receptor into the previously described functional categories by comparing the expression in the resting state versus after activation in the available data from different cell types (*Rumpret et al., 2020*; *Table 3*, *Supplementary file 2*). We defined negative feedback receptors as below median expression in the resting state, but expressed after activation. Threshold receptors were defined as being expressed in the resting state and having less than 0.5 log2 fold change after activation, while threshold-disinhibition and threshold-negative feedback receptors were defined by more than 0.5 log2 fold decrease or increase in expression upon activation, respectively. Using this functional classification system, we found that all immune cell subsets, except B cells, expressed receptors of all four categories, although all subsets primarily expressed threshold receptors.

We did not observe many changes in inhibitory receptor gene expression after activation in neutrophils. Six receptor genes, such as *SLAMF1*, were absent on resting neutrophils, but expression was induced after stimulation, categorising them as negative feedback receptors. We identified 124 receptors as threshold receptors, while four and three genes were identified as threshold-disinhibition and threshold-negative feedback receptors, respectively (*Figure 2B*).

In contrast, in monocytes 26 genes, including *SIGLEC8* or *SIGLEC11*, had the expression pattern of negative feedback receptors. We found 72 receptors, e.g., *SIGLEC10* or *PILRA*, that maintained their expression after activation, while 21 were downregulated, such as *VSTM1* or *CLEC12A*. Only a single receptor, *SLAMF7*, was categorised as a threshold-negative feedback receptor. Of note, a cluster of 14 receptors was shared exclusively between monocytes and neutrophils (*Figure 2C*) and thus could be indicative of a set of conserved myeloid-specific checkpoints.

We observed the expression of 17 negative feedback receptors, e.g., *SIGLEC15* and *VSTM4*, and 81 threshold receptors, such as *CD244* and *CD300A* on NK cells. Additionally, 11 genes were classified as threshold-disinhibition receptors, e.g., *FCRL6*, while 2 were determined to be threshold-negative feedback receptors.

B cells expressed 111 predicted inhibitory receptor genes, of which 10, like *SIRPA* and *SIGLEC6*, were negative feedback receptors. We found 85 threshold receptors, such as *BTLA*, as well as 16 threshold-disinhibition receptors, e.g., *PILRA* and *LST1*. B cells were the only subset that are not predicted to express threshold-negative feedback receptors. B cells also shared expression of six receptors with monocytes and neutrophils, while two genes were uniquely expressed by B cells.

CD4[+] and CD8[+] T cells expressed 8 and 9 negative feedback receptors, respectively, such as *PDCD1*, in addition to 73 (on CD4[+] T cells) and 78 (on CD8[+] T cells) threshold receptors, e.g., *CEACAM1* and *SIT1*. In addition, we found 16 threshold-disinhibition receptors on CD4[+] T cells and 15 on CD8[+] T cells, such as *LAIR1* on CD4[+] T cells and *AXL* on both subsets, as well as 2 threshold-negative feedback receptors on CD4[+] and 4 on CD8[+] T cells.

All immune cell subsets expressed fewer multi-membrane spanning inhibitory receptor genes compared to single spanners (*Figure 2—figure supplement 1*, *Supplementary file 2*). When we assigned these multi-spanning targets to the different functional categories, a similar picture emerged as for the single spanners, where most receptors categorised as threshold receptors in all immune cell types. Whereas monocytes, B cells, CD4[+] and CD8[+] T cells expressed more single-spanning threshold-disinhibition receptors than negative feedback receptors, the opposite was observed

**Table 3.** Number of single-spanning receptors in different functional categories for each immune cell subset.

| Cell type | Not expressed | Negative feedback | Threshold-negative feedback | Threshold-disinhibition | Threshold |
|---|---|---|---|---|---|
| Neutrophils | 78 | 6 | 3 | 4 | 124 |
| Monocytes | 95 | 26 | 1 | 21 | 72 |
| NK cells | 104 | 17 | 2 | 11 | 81 |
| B cells | 104 | 10 | 0 | 16 | 85 |
| CD4 T cells | 116 | 8 | 2 | 16 | 73 |
| CD8 T cells | 109 | 9 | 4 | 15 | 78 |

for multi-spanning inhibitory receptor genes. NK cells had less dynamic gene regulation of multi-spanning targets than of single spanners, while categorisation for neutrophils was similar between multi- and single spanners (*Figure 2—figure supplement 1B*). Interestingly, the multi-spanning targets were shared between three or more subsets more often than was the case for the single spanners (*Figure 2—figure supplement 1C*).

In summary, we identified 398 known and putative inhibitory receptors being expressed across a variety of immune cell subsets, with mostly overlapping expression patterns. We classified these receptors into different functional categories based on their expression in the resting and activated states. We found that all immune cells expressed mostly threshold receptors. Among the receptors that changed expression after activation, monocytes and NK cells mainly expressed negative feedback receptors, while the T cells and B cells expressed more threshold-disinhibition receptors. Neutrophils did not have very dynamic gene expression patterns.

## Known and putative inhibitory receptors are expressed on tumour infiltrating T cells

To further translate our findings to a disease context, we explored the expression of putative inhibitory receptor genes in tumour infiltrating lymphocytes. We used publicly available single-cell RNA sequencing data of CD4$^+$ and CD8$^+$ T cells from 21 types of cancer and determined inhibitory receptor gene expression in the different T cell subsets identified by *Zheng et al., 2021b*; *Figure 3A*. After quality control and filtering, the expression of 133/215 single spanners and 111/197 of the multi-spanners could be assessed in this dataset.

Among T cell subsets, expression of single-spanning inhibitory receptor genes was variable (*Figure 3A*). We found 27 and 46 inhibitory receptor genes in naïve CD4$^+$ and CD8$^+$ T cells, respectively, while T cells with a more differentiated phenotype, e.g., regulatory CD4$^+$ T cells (68 genes) and exhausted CD8$^+$ T cells (83 genes), expressed a wider array of known and predicted inhibitory receptor genes (*Figure 3B and C*).

When we assessed expression of multi-spanning inhibitory receptor genes (*Figure 3—figure supplement 1A*), we found similar variable expression across T cell subsets as for the single spanners. Naïve CD4$^+$ T cells expressed 21 multi-spanning receptor genes, while many more were found in the other subsets (*Figure 3—figure supplement 1B*). Resident memory CD8$^+$ T cells expressed 87 multi-spanning inhibitory receptor genes, while CD8$^+$ memory and effector memory expressed 9 and 26 genes, respectively (*Figure 3—figure supplement 1C*).

Based on the previously proposed functional categorisation of known and novel inhibitory receptors (*Figure 2* and *Figure 2—figure supplement 1*), we assessed the functional categories expressed across the tumour infiltrating T cell subsets (*Figure 3—figure supplement 2*). Similarly to the immune subset analysis, we observed that the subsets primarily expressed threshold receptors.

To validate that we rediscovered known immunotherapeutic targets, we assessed the expression of the known inhibitory receptor genes in tumour infiltrating lymphocytes of melanoma patients using the same dataset (*Zheng et al., 2021b*). We found high expression of known immunotherapeutic targets, such as PD-1, in addition to other inhibitory receptors being targeted in clinical trials, such as TIGIT (*Figure 3—figure supplement 3*).

Overall, this shows that the putative inhibitory immune receptors we identified are expressed on tumour infiltrating T cells and differ between cellular subsets. Further investigation into differences and similarities between these populations in different cancer types could assist in targeting these receptors for therapy.

In conclusion, we designed a novel, combined bioinformatics approach to predict around 400 immune inhibitory receptors in the human genome, including both single- and multi-spanning targets. Our list of novel targets will assist drug target selection in diseases where inhibitory receptor targeting is warranted.

## Discussion

This study introduces some major improvements over previous attempts at predicting novel inhibitory receptors in the human genome. In 2004, Staub et al. performed a search limited to type I membrane receptors containing an annotated extracellular domain and a restricted ITIM sequence.

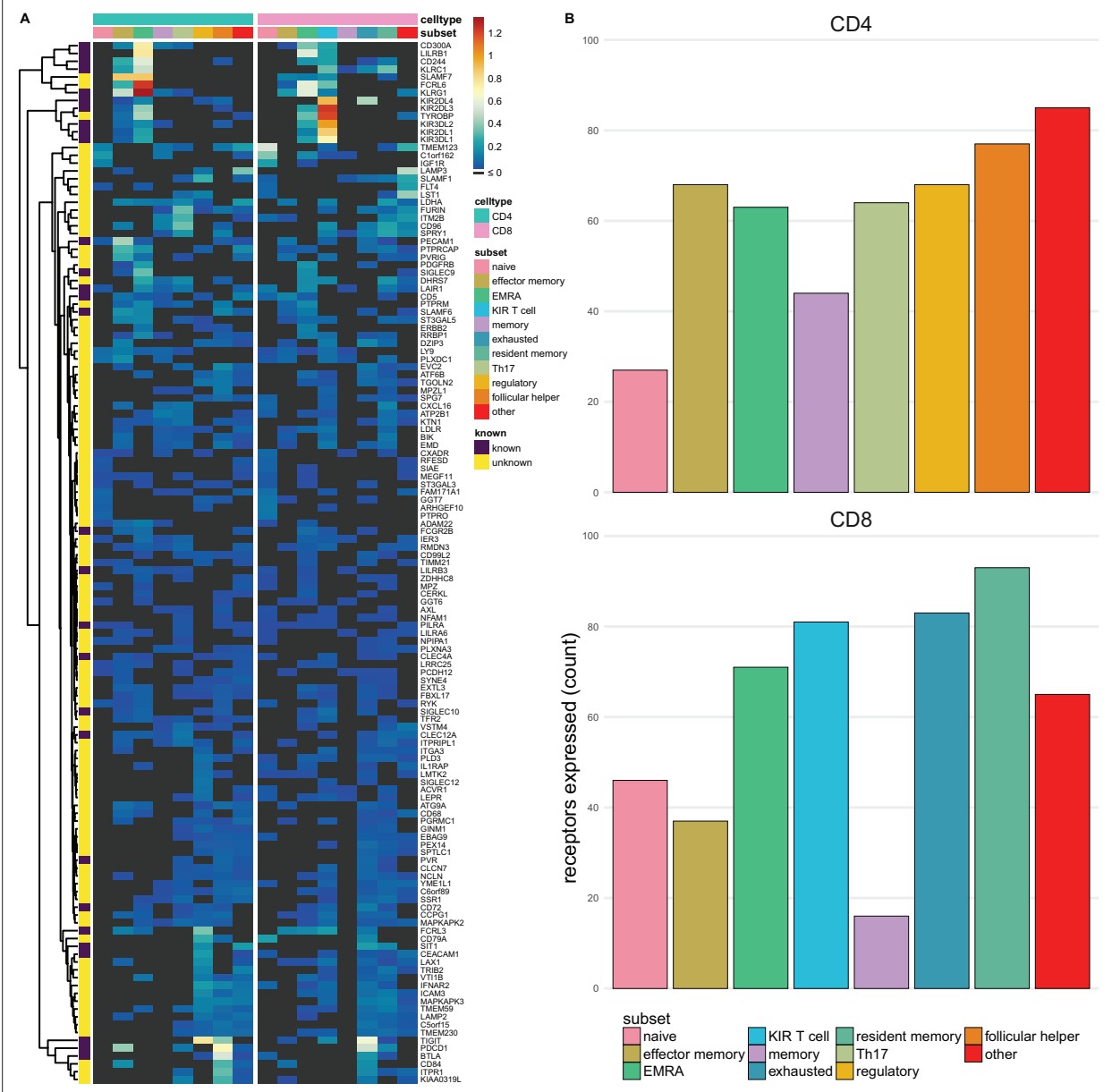

**Figure 3.** Single-spanning predicted inhibitory receptors are expressed across a wide variety of tumour infiltrating T cell subsets. (**A**) Heatmap with row-normalised expression data for known and predicted single-spanning inhibitory receptor genes in different tumour infiltrating T cell subsets. Receptors were considered not expressed in a T cell subset (black) when expression was below median across all subsets. (**B**) Number of inhibitory receptor genes expressed by different CD4[+] T cell subsets (upper graph) and CD8[+] T cell subsets (lower graph).

The online version of this article includes the following figure supplement(s) for figure 3:

**Figure supplement 1.** Multi-spanning predicted inhibitory receptors are expressed across a wide variety of tumour infiltrating T cell subsets.

**Figure supplement 2.** Expression of functional categories of inhibitory receptors across tumour infiltrating T cell subsets.

**Figure supplement 3.** Known inhibitory receptors are expressed in tumour infiltrating T cell subsets of melanoma patients.

They identified a total of 94 genes encoding inhibitory receptors, of which 32 had been previously described to rely on their ITIM to inhibit immune function (*Staub et al., 2004*). A later study by *Daëron et al., 2008*, performed a more extensive analysis of ITIM-bearing receptors across multiple species, including chickens and *Drosophila* (*Daëron et al., 2008*). In humans, they identified 613 predicted type I and type II single-spanners and 992 multi-spanning transmembrane proteins, corresponding to around 1500 genes. This number agrees with what we retrieved in the initial steps of

our bioinformatics pipeline, where we identified 1562 genes. Our efforts to reduce this number to candidates with a high probability of functioning as inhibitory receptors resulted in 390 novel targets. Compared to the previous studies, our method has key advantages. Firstly, the TOPCONS algorithm we used is superior to TMHMM and Philius, which have been used previously, at predicting membrane topology (*Tsirigos et al., 2015*; *Möller et al., 2001*; *Reynolds et al., 2008*). Most importantly, we excluded many potential false positives by filtering out proteins that are likely to contain an ITIM by chance and used three-dimensional structure predictions to enrich the remaining targets for those containing ITIMs in unstructured regions. Furthermore, our expression analysis and functional categorisation of the putative inhibitory receptors can assist in gaining a better understanding of their relevance in different contexts.

An important component of our pipeline is the use of AlphaFold (*Jumper et al., 2021*). Our approach was exceptionally good at filtering out many proteins in the olfactory receptor family, for which almost all ITIMs identified were found in highly structured regions. We determined a threshold of 80 pLDDT based on the calculated prediction scores of the ITIM/ITSMs in known inhibitory receptors, as this encompassed 99 out of 101 ITIM/ITSMs. However, this excluded ITIMs in *CD5* (Y402) and *BTLA* (Y282), with confidence scores of 85.2 and 81.5, respectively. Of note, studies have suggested that this particular ITIM in *CD5* might not be functional or essential for CD5 function (*Voisinne et al., 2018*). It is important to highlight that the AlphaFold structure predictions are based on the proteins being in isolation. It is possible that ITIMs become available after conformational changes following ligand binding, or that ITIMs in disordered regions are not functional when the proteins are found in complexes (*Jakob et al., 2014*; *Kiefhaber et al., 2012*).

We acknowledge that our approach has limitations. We used known ITIM-bearing inhibitory receptors as validation for the various steps of the pipeline and retrieved all but one receptor, resulting in a retrieval of 98%. We did not retrieve *MPIG6B*, as we were not able to properly predict the transmembrane helix in the ITIM-containing isoforms of *MPIG6B*. With increasing computing power and further development of artificial intelligence, these prediction algorithms are likely to further improve (*Rives et al., 2021*; *Tsirigos et al., 2018*). The identification of novel ITIM/ITSM-bearing inhibitory receptors also relies on our current knowledge of known inhibitory receptors and their motifs. This influences the thresholds in our analyses, and might be skewed by larger protein families, e.g., KIR- and LILRB-family receptors. Additional filtering of the candidates could be achieved by assessing subcellular localisation, as it is possible that the transmembrane proteins we identified are not localised on the outer membrane, but instead on intracellular membranes. Although they could still function as inhibitory receptors, this would complicate the ability to effectively target them with therapeutic antibodies, and might instead require other blocking agents, such as proteolysis-targeting chimeras (PROTACS) or small molecule inhibitors (*Békés et al., 2022*).

We focused our search on receptors bearing ITIM or ITSM sequences, while inhibitory receptors have been described that function independent of these short sequences. We previously reported that CD200R has a unique signalling motif that is not shared with any other protein (*Timmerman et al., 2021*). Also, TIM-3 contains multiple tyrosine residues that are not part of any known signalling motif (*Wolf et al., 2020*). A small number (26 genes) of putative inhibitory receptors contained an ITIM embedded in an ITAM sequence, i.e., $Yxx(I|L)x_{6-12}Yxx(I|L)$. Although some ITAM-bearing molecules are capable of mediating inhibitory signals in an ITIM-independent manner, these receptors may not all be inhibitory (*Barrow and Trowsdale, 2006*).

Until now, no known functional multimembrane spanning immune inhibitory receptors have been reported, although some ITIM-bearing multi-spanners, e.g., *SSTR2* and *HCRTR1*, were shown to recruit SHP-2 (*Ferjoux et al., 2003*; *Voisin et al., 2008*). It is possible that multi-spanning inhibitory receptors have not yet been fully characterised due to their size and limited targetability, making them difficult to study. Of interest, in our analysis multi-spanning receptors were often expressed by multiple immune cell subsets. This may suggest that multi-spanning inhibitory receptors would be capable of inhibiting a broad array of immune responses.

We previously argued how different functional categories of inhibitory receptors might serve distinct purposes (*Rumpret et al., 2020*). Here, we grouped known and putative inhibitory receptors in different functional categories based on their expression in isolated immune cell subsets in resting and stimulated state and show that a large number of them is expressed and regulated in a cell type-specific manner. However, as we categorised these receptors based on in vitro activation with limited

stimuli, expression must be further validated in in vivo settings. Nonetheless, we tried to characterise the functional categories expressed by tumour infiltrating lymphocytes by extrapolating the in vitro defined functional categorisation per gene. This showed that mainly threshold receptors and some (threshold-)negative feedback receptors are expressed by the different T cell subsets. This would open the possibility of targeting different functional categories for cancer immunotherapy, since targeting a threshold receptor to lower the threshold for activation and a negative feedback receptor to lengthen and strengthen the cellular response might therefore be more effective than targeting two receptors of a single functional category. However, we acknowledge that this will require further validation of expression patterns in vivo in different cancers and immune cell subsets.

We found that the highest number of target genes are expressed in neutrophils, which fits the need to control and precisely regulate their potentially damaging effector functions (*Geerdink et al., 2018*). However, inhibitory receptor gene expression on these cells did not change much after activation. This does not exclude regulation of receptor expression at the protein level, which would fit with the more short-lived nature of these cells. For instance, LAIR-1 protein is stored in intracellular granules which results in rapid membrane expression on activated granulocytes (*Geerdink et al., 2018*). On monocytes, we found many negative feedback receptors. Considering that monocytes (especially macrophages) play a central role in orchestrating the overall immune response, timely resolution of their activation is key to successful resolution of inflammation, which could be achieved by these receptors.

For the functional categorisation of novel inhibitory receptor genes, we collected multiple publicly available datasets containing different isolated immune cell subsets in the resting state and after cell type-specific activation. For 115 genes, we predicted a non-canonical isoform, but not the canonical sequence, as a putative inhibitory receptor. This means that our expression analyses for these inhibitory receptor genes might not be fully reflective of the expression of individual isoforms of those genes, considering that isoform usage can vary widely between different cells and activation states, as has been well described for CEACAM1 (*Gaur et al., 2008*; *Helfrich and Singer, 2019*). In the present study, we also did not consider the ligand expression of these known and putative receptors. Regulation of inhibitory receptors does not only occur at the level of the receptor itself, but the presence or absence of ligands, decoy receptors, and other interactions all add additional layers of regulation of control to the immune system (*Rumpret et al., 2020*).

To further prioritise inhibitory receptors in immune cell subsets or diseases of interest, gene co-expression networks of putative inhibitory receptors could be assessed. On the one hand, the co-occurrence of putative inhibitory receptors with known inhibitory receptors within a module could be one approach, while on the other hand the presence of putative inhibitory receptors in a different module could suggest novel regulation of different biological functions than the known receptors. The location of the putative inhibitory receptors in the network could also change depending on the cell type and the activation status of the cell. Additionally, one could look at the co-expression of candidates with other genes within a gene module to look at potential biological function, and at co-expression with signalling molecules known to interact with inhibitory receptors, such as Csk, SHP-1, SHP-2, and SHIP1, although their regulation might be more post-translationally regulated rather than at the transcriptional level.

As proof-of-concept, we assessed expression of both known and novel targets in various tumour infiltrating T cell subsets. This helps understanding how these potential drug targets are differentially expressed in a disease context. As many receptors we identified were shared across immune cell subsets, it would be particularly interesting to block those that are enriched in, e.g., effector CD8+ T cell subsets. While some T cell subsets, such as exhausted CD8+ T cells and CD4+ regulatory T cells, appear to not differ much in their expression of either single- or multi-spanning receptors, we do observe that, for example, effector memory CD4+ T cells or EMRA CD8+ T cells express single-spanning inhibitory receptors to a higher extent than multi-spanning inhibitory receptors. It is possible that these differences and similarities reflect some of the roles multi-spanning inhibitory receptors could play in regulating immune cells, e.g. in response to chemokines, as many chemokine receptors are multi-spanning proteins. In contrast, receptors expressed in subsets that can have pro-tumourigenic properties, such as CD4+ T$_{reg}$ or Th17 cells (*Tay et al., 2021*; *Karpisheh et al., 2022*), should not be activated; these targets could instead benefit from agonist therapy. Furthermore, these putative receptors might also be attractive in autoimmune diseases, where immune checkpoints are

increasingly being explored as potential targets for therapy (*Grebinoski and Vignali, 2020*; *Zhai et al., 2021*).

Taken together, we have designed a bioinformatic pipeline to identify a key list of 390 predicted inhibitory receptors to be used as a starting point for further research into novel targets for targeting in diseases.

## Methods

### Identification of putative inhibitory receptors in the human genome

To identify putative inhibitory receptors in the human genome, amino acid sequences for 96,457 protein coding, as annotated by Ensembl, transcripts were retrieved from Ensembl (release 105, December 2021) (*Cunningham et al., 2022*) using the *biomaRt* package (*Durinck et al., 2009*) in R. Amino acid sequences were then filtered based on the presence of an ITIM or ITSM sequence, (V|L|I|S|T)xYxx(I|L|V). Seven out of 40,063 ITIM-containing proteins were larger than 10,000 amino acids, and were excluded because they are not feasible for membrane topology prediction using the TOPCONS method (*Tsirigos et al., 2015*). Topology results for the remaining 40,056 proteins were then retrieved, and proteins were further filtered based on the presence of a predicted transmembrane domain (12,576 out of 40,056 proteins). For the remaining proteins, we determined whether the identified ITIM or ITSM sequence corresponded to an intracellular section of the protein, which resulted in 4613 protein sequences remaining, corresponding to 1562 individual genes (*Figure 1A*). A further 61 protein sequences were excluded as they could not be matched to a HUGO Gene Nomenclature Committee (HGNC) gene symbol (*Seal et al., 2023*).

### Determining the likelihood of intracellular protein domain containing an ITIM by chance

To determine the odds of finding an intracellular ITIM or ITSM sequence in a specific protein, all intracellular domains of the proteins were retrieved based on the TOPCONS predictions. For each protein, the amino acids in the intracellular domain were then shuffled randomly, using the *stringi* package (*Gagolewski, 2022*), and the number of identified ITIM or ITSM sequences in the permutated sequences was compared to the actual number of intracellular ITIM or ITSM sequences in the original protein. Each protein was permutated 10,000 times, and the number of occurrences of a permutated protein containing more or equal ITIM/ITSMs to the original sequence was counted and used as measure for the likelihood of finding an ITIM or ITSM by chance. A threshold was defined based on likelihood scores for the proteins corresponding to known inhibitory receptors, and was set at 0.25+ε, with ε being a random number determined error between 0 and 0.01.

### Determining AlphaFold model confidence scores for individual ITIMs

Three-dimensional structure of protein sequences that passed the likelihood threshold were next determined using AlphaFold (v2.2.0) under default parameters with amber minimisation (run_relax = True) using the pdb70, uniref90, mgnify (mgy_clusters_2018_12), uniclust30 (uniclust30_2018_08), and bfd (bfd_metaclust_clu_complete_id30_c90_final_seq) databases on a high-performance cluster (*Jumper et al., 2021*; *Varadi et al., 2022*). AlphaFold was not able to determine the structure for 46 of these proteins, of which 6 were too long and 40 returned a run time error involving HHblits and the hh-suite. The three-dimensional prediction of 37 of these proteins was instead determined using ColabFold (*Mirdita et al., 2022*) (v1.5.2) using MMseqs2 and HHsearch under default parameters without using template information (template_mode = none) and using mmseq2_uniref_env as MSA mode. The top ranked prediction files were retrieved for each protein, and loaded into R using the *bio3d* package (*Grant et al., 2021*). Prediction confidence scores (pLDDT) were retrieved, and the average pLDDT for all six amino acids encompassing the ITIM or ITSM sequence was used as the AlphaFold confidence score for that ITIM or ITSM. Proteins that had no ITIM or ITSM sequences with an average confidence score below 80 were not used in further analysis.

### Classification of receptor functional categories

Based on the in vitro RNA sequencing dataset of immune cells in the resting state and after activation, functional categories were defined based on expression patterns as outlined previously

(*Rumpret et al., 2020*). Data was retrieved from *Calderon et al., 2019*, and *West et al., 2016*. Gene symbols were matched to Ensembl Gene IDs using biomaRt and Ensembl (release 108, October 2022) (*Cunningham et al., 2022*). For the neutrophil data (*West et al., 2016*), TPM values were retrieved and log2 scaled. For other immune cells, count matrix was normalised using the *DESeq2* package (*Love et al., 2014*). Culture conditions for all immune cells were simplified to major immune cell subsets, and to *resting* and *stimulated* (*Table 2*). Average expression for each inhibitory receptor gene was determined for each sample. An inhibitory receptor gene was considered expressed when it was above the median expression of all genes within the sample. Genes that were not expressed in the resting state, but were expressed in the stimulated state, were considered negative feedback receptors. Genes expressed in the resting state were considered threshold, threshold-disinhibition, or threshold-negative feedback receptors if the log2 fold change in expression was between –0.5 and 0.5, below –0.5, or above 0.5, respectively. Genes expressed in the resting state, but not after activation, were also considered threshold-disinhibition receptors.

## Collection and normalisation of immune cell RNA sequencing data

For the analysis of tumour infiltrating T cells, a normalised expression matrix was retrieved from *Zheng et al., 2021b*; *Zheng and Qin, 2021a*. The average Z score of all cells in a T cell subset was determined, weighted by the relative contribution of different subclusters identified in the study. Inhibitory receptor genes were considered expressed in a T cell subset if the average Z score was above 0.

## Acknowledgements

We would like to thank the IT team at ETHZ for their assistance with the setup of AlphaFold and the members of the Inhibitory Receptor Lab for their helpful comments during the preparation of the manuscript.

## Additional information

### Competing interests

Akashdip Singh: AS is named as co-inventor on a patent application (patent application number: PCT/NL2024/050501) based on this work. Alberto Miranda Bedate: AMB is named as co-inventor on a patent application (patent application number: PCT/NL2024/050501) based on this work. Helen J von Richthofen: HJR is named as co-inventor on a patent application (patent application number: PCT/NL2024/050501) based on this work. Michiel van der Vlist: MV is named as co-inventor on a patent application (patent application number: PCT/NL2024/050501) based on this work. Jürgen J Kuball: JK is cofounder, shareholder and scientific advisor of GADETA. JK is named as co-inventor on a patent application (patent application number: PCT/NL2024/050501) based on this work. Can Kesmir: CK is named as co-inventor on a patent application (patent application number: PCT/NL2024/050501) based on this work. M Ines Pascoal Ramos: IR is named as co-inventor on a patent application (patent application number: PCT/NL2024/050501) based on this work. Linde Meyaard: LM has received funding for investigator-initiated studies on inhibitory receptors from Boehringer Ingelheim, Next-Cure, NGM Biopharmaceuticals and Argenx. LM is named as co-inventor on a patent application (patent application number: PCT/NL2024/050501) based on this work. The other authors declare that no competing interests exist.

### Funding

| Funder | Grant reference number | Author |
| --- | --- | --- |
| Oncode Institute | | Akashdip Singh<br>Helen J von Richthofen<br>Saskia V Vijver<br>Michiel van der Vlist<br>M Ines Pascoal Ramos<br>Linde Meyaard |
| KWF Kankerbestrijding | 2021-14339 | Linde Meyaard |

| Funder | Grant reference number | Author |
| --- | --- | --- |

The funders had no role in study design, data collection and interpretation, or the decision to submit the work for publication.

## Author contributions

Akashdip Singh, Data curation, Formal analysis, Investigation, Visualization, Writing – original draft, Writing – review and editing; Alberto Miranda Bedate, Conceptualization, Data curation, Software, Formal analysis, Investigation, Methodology; Helen J von Richthofen, Conceptualization, Methodology; Saskia V Vijver, Formal analysis, Investigation, Visualization, Methodology, Writing – review and editing; Michiel van der Vlist, Conceptualization, Methodology, Writing – review and editing; Raphael Kuhn, Resources, Data curation, Investigation, Methodology; Alexander Yermanos, Data curation, Supervision, Investigation, Methodology, Writing – review and editing; Jürgen J Kuball, Supervision, Funding acquisition, Project administration, Writing – review and editing; Can Kesmir, Conceptualization, Supervision, Methodology, Writing – review and editing; M Ines Pascoal Ramos, Conceptualization, Supervision, Methodology, Writing – original draft, Writing – review and editing; Linde Meyaard, Conceptualization, Supervision, Methodology, Writing – original draft, Project administration, Writing – review and editing

## Author ORCIDs

Akashdip Singh (ID) https://orcid.org/0000-0001-5326-8826
Saskia V Vijver (ID) http://orcid.org/0000-0003-2526-3045
Linde Meyaard (ID) https://orcid.org/0000-0003-0707-4793

Reviewer #2 (Public review): https://doi.org/10.7554/eLife.92870.3.sa1
Author response https://doi.org/10.7554/eLife.92870.3.sa2

## Additional files

### Supplementary files

• Supplementary file 1. Overview of predicted immune inhibitory receptor genes bearing an intracellular immunoreceptor tyrosine-based inhibitory motif (ITIM). Supplementary tables containing all genes and corresponding peptides containing an ITIM in the intracellular domain as determined through the steps described in *Figure 1*, further subdivided into genes that additionally pass the likelihood and AlphaFold filters (Table 1a), those that pass the likelihood, but not AlphaFold filter (Table 1b) and those that did not pass the likelihood, nor the AlphaFold filter (Table 1c).

• Supplementary file 2. Number of multi-spanning receptors in different functional categories for each immune cell subset. Supplementary table corresponding to *Figure 2—figure supplement 1* contains the numbers of multi-spanning receptors that have been categorised as 'not expressed', 'negative feedback', 'threshold-negative feedback', 'threshold-disinhibition', and 'threshold' in different immune cell subsets, i.e., neutrophils, monocytes, NK cells, B cells, and $CD4^+$ and $CD8^+$ T cells.

• MDAR checklist

### Data availability

Code and data used to generate the figures in this study can be retrieved from https://github.com/AkashdipSingh/IIR_pipeline (copy archived at *AkashdipSingh, 2023*).

The following previously published datasets were used:

| Author(s) | Year | Dataset title | Dataset URL | Database and Identifier |
|---|---|---|---|---|
| Zheng L, Qin S, Si W, Wang A, Xing B, Gao R, Ren X, Wang L, Wu X, Zhang J, Wu N, Zhang N, Zheng H, Ouyang H, Chen K, Bu Z, Hu X, Ji J, Zhang Z | 2021 | Pan-Cancer Single Cell Landscape of Tumor-Infiltrating T Cells | https://www.ncbi.nlm.nih.gov/geo/query/acc.cgi?acc=GSE156728 | NCBI Gene Expression Omnibus, GSE156728 |
| West EE, Spolski R, Kazemian M, Zx Yu, Kemper C, Leonard WJ | 2016 | TSLP acts on neutrophils to drive complement-mediated killing of methicillin-resistant *Staphylococcus aureus* | https://www.ncbi.nlm.nih.gov/geo/query/acc.cgi?acc=GSE73313 | NCBI Gene Expression Omnibus, GSE73313 |
| Calderon D, Nguyen ML, Mezger A | 2019 | RNA-seq data | https://www.ncbi.nlm.nih.gov/geo/query/acc.cgi?acc=GSE118165 | NCBI Gene Expression Omnibus, GSE118165 |

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
