## [Editor Report · eLife assessment]

The authors presented a **valuable** bioinformatics pipeline for screening and identifying inhibitory receptors for potential drug targets. They provided **solid** evidence showing a sequential reduction in the search space through various screening tools and algorithms and demonstrated that this pipeline can be used to "rediscover" known targets. Further experimental validation on putative and unknown inhibitory receptors will strengthen the evidence reported in this work. This study will be of interest to bioinformaticians and computational biologists working on immune regulation, sequence screening, and target identification of immune checkpoint inhibitors.

---

## [Referee Report · Reviewer #2 (Public review)]

Summary:

The authors developed a bioinformatic pipeline to aid the screening and identification of inhibitory receptors suitable as drug targets. The challenge lies in the large search space and lack of tools for assessing the likelihood of their inhibitory function. To make progress, the authors used a consensus protein membrane topology and sequence motif prediction tool (TOPCOS) combined with both a statistical measure assessing their likelihood function and a machine learning protein structural prediction model (AlphaFold) to greatly cut down the search space. After obtaining a manageable set of 398 high confidence known and putative inhibitory receptors through this pipeline, the authors then mapped these receptors to different functional categories across different cell types based on their expression both in the resting and activated state. Additionally, by using publicly available pan cancer scRNA-seq for tumor-infiltrating T cells data, they showed that these receptors are expressed across various cellular subsets.

Strengths:

The authors presented sound arguments motivating the need to efficiently screen inhibitory receptors and to identify those that are functional. Key components of the algorithm were presented along with solid justification for why they addressed challenges faced by existing approaches. To name a few:

• TOPCON algorithm was elected to optimize the prediction of membrane topology

• A statistical measure was used to remove potential false positives

• AlphaFold is used to filter out putative receptors that are low confidence (and likely intrinsically disordered)

To examine receptors screened through this pipeline through a functional lens, the authors proposed to look at their expression of various immune cell subsets to assign functional categories. This is a reasonable and appropriate first step for interpreting and understanding how potential drug targets are differentially expressed in some disease contexts. They also presented an example showing this pipeline can be used to "rediscover" known targets.

Weaknesses:

The paper has strength in the pipeline they presented, but the weakness, in my opinion, lies in the lack of direct experimental validation on putative receptors. That said, the authors presented in the revised manuscript, as a proof-of-concept, an analytic approach for using functional categorization of putative inhibitory receptors to select therapeutic targets based on in vitro RNAseq. Such analysis will benefit from further investigation across different cancer types using in vivo expression.

---

## [Author Response]

The following is the authors’ response to the original reviews.

**eLife assessment**
This work is potentially useful because it has generated a mineable yield of new candidate immune inhibitory receptors, which can serve both as drug targets and as subjects for further biological investigation. It is noted however that the argument of the work is rather incomplete, in that it does very little to validate the putative new receptors, and merely makes a study of their putative distribution across cell types. Experimental follow-up to demonstrate the claimed properties for the proteins identified, or mining existing experimental data sources on gene expression across tissues to at least show that the pipeline correctly identified genes likely to be specific to immune cells (or something along these lines), would make this work more complete and compelling.

We thank the editors for their critical reading and assessment of our manuscript. We acknowledge that the present study is limited by a lack of experimental follow-up. However, we purposely chose to make this pipeline of putative novel inhibitory receptors public at this early stage for our work to be a starting point for further functional investigation of these targets by the scientific community.

**Public Reviews:**

**Reviewer #1 (Public Review):**
This manuscript proposes a new bioinformatics approach identifying several hundreds of previously unknown inhibitory immunoreceptors. When expressed in immune cells (such as neutrophils, monocytes, CD8+, CD4+, and T-cells), such receptors inhibit the functional activity of these cells. Blocking inhibitory receptors represents a promising therapeutic strategy for cancer treatment.As such, this is a high-quality and important bioinformatics study. One general concern is the absence of direct experimental validation of the results. In addition to the fact that the authors bioinformatically identified 51 known receptors, providing such experimental evaluation (of at least one, or better few identified receptors) would, in my opinion, significantly strengthen the presented evidence.I will now briefly summarize the results and give my comments.First, using sequence comparison analysis, the authors identify a large set of putative receptors based on the presence of immunoreceptor tyrosine-based inhibitory motifs (ITIMs), or immunoreceptor tyrosinebased switch motifs (ITSMs). They further filter the identified set of receptors for the presence of the ITIMs or ITSMs in an intracellular domain of the protein. Second, using AlphaFold structure modeling, the authors select only receptors containing ITIMs/ITSMs in structurally disordered regions. Third, the evaluation of gene expression profiles of known and putative receptors in several immune cell types was performed. Fourth, the authors classified putative receptors into functional categories, such as negative feedback receptors, threshold receptors, threshold disinhibition, and threshold-negative feedback. The latter classification was based on the available data from Nat Rev Immunol 2020. Fifth, using publicly available single-cell RNA sequencing data of tumor-infiltrating CD4+ and CD8+ cells from nearly twenty types of cancer, the authors demonstrate that a significant fraction of putative receptors are indeed expressed in these datasets.In summary, in my opinion, this is an interesting, important, high-quality bioinformatics work. The manuscript is clearly written and all technical details are carefully explained.One comment/suggestion regarding the methodology of evaluating gene expression profiles of putative receptors: perhaps it might be important to look at clusters of genes that are co-expressed with putative inhibitory receptors.

We thank the reviewer for their comments and suggestions. We acknowledge that looking at co-expressed genes and subsequently at gene ontology enrichment could be an interesting approach to prioritize the inhibitory receptors. However, since there are many ways to approach the results of the gene coexpression networks, which also depend on the cell type and activation status of interest, we have chosen to discuss the implications of these networks in the discussion with the following paragraph, rather than reporting all these different approaches in the paper:

“To further prioritize inhibitory receptors in immune cell subsets or diseases of interest, gene coexpression networks of putative inhibitory receptors could be assessed. On the one hand, the cooccurrence of putative inhibitory receptors with known inhibitory receptors within a module could be one approach, while on the other hand the presence of putative inhibitory receptors in a different module could suggest novel regulation of different biological functions than the known receptors. The location of the putative inhibitory receptors in the network could also change depending on the cell type and the activation status of the cell. Additionally, one could look at the co-expression of candidates with other genes within a gene module to look at potential biological function, and at co-expression with signalling molecules known to interact with inhibitory receptors, such as Csk, SHP-1, SHP-2 and SHIP1, although their regulation might be more post-translationally regulated rather than at mRNA level.”

**Reviewer #2 (Public Review):**
Summary:The authors developed a bioinformatic pipeline to aid the screening and identification of inhibitory receptors suitable as drug targets. The challenge lies in the large search space and lack of tools for assessing the likelihood of their inhibitory function. To make progress, the authors used a consensus protein membrane topology and sequence motif prediction tool (TOPCOS) combined with both a statistical measure assessing their likelihood function and a machine learning protein structural prediction model (AlphaFold) to greatly cut down the search space. After obtaining a manageable set of 398 high-confidence known and putative inhibitory receptors through this pipeline, the authors then mapped these receptors to different functional categories across different cell types based on their expression both in the resting and activated state. Additionally, by using publicly available pan-cancer scRNA-seq for tumor-infiltrating T-cell data, they showed that these receptors are expressed across various cellular subsets.Strengths:The authors presented sound arguments motivating the need to efficiently screen inhibitory receptors and to identify those that are functional. Key components of the algorithm were presented along with solid justification for why they addressed challenges faced by existing approaches. To name a few:• TOPCON algorithm was elected to optimize the prediction of membrane topology.• A statistical measure was used to remove potential false positives.• AlphaFold is used to filter out putative receptors that are low confidence (and likely intrinsically disordered).To examine receptors screened through this pipeline through a functional lens, the authors proposed to look at their expression of various immune cell subsets to assign functional categories. This is a reasonable and appropriate first step for interpreting and understanding how potential drug targets are differentially expressed in some disease contexts.Weaknesses:The paper has strength in the pipeline they presented, but the weakness, in my opinion, lies in the lack of concrete demonstration on how this pipeline can be used to at least "rediscover" known targets in adisease-specific manner. For example, the result that both known and putative immune inhibitory receptors are expressed across a wide variety of tumor-infiltrating T-cell subsets is reassuring, but this would have been more informative and illustrative if the authors could demonstrate using a disease with known targets, as opposed to a pan-cancer context. Additionally, a discussion that contrasts the known and putative receptors in the context above would help readers better identify use cases suitable for their research using this pipeline. Particularly,• For known receptors, does the pipeline and the expression analysis above rediscover the known target in the disease of interest?• For putative receptors, what do the functional category mapping and the differential expression across various tumor-infiltrating T-cell subsets imply on a potential therapeutic target?

We thank the reviewer for their assessment and comments. The primary purpose of the bioinformatics pipeline was to identify putative inhibitory receptors in a disease-agnostic manner and allow the scientific community to further explore targets in their specific diseases of interest. We performed our pan-cancer expression analysis as a preliminary proof of concept and agree that exploring targets in specific diseases, cancer or otherwise, could be more informative. To validate that we rediscovered known immunotherapeutic targets, we analyzed the expression of known inhibitory receptors on tumorinfiltrating T cells of melanoma patients using the same dataset as figure 3. We find high expression of known therapeutic targets, such as PD-1, in addition to other known inhibitory receptors that are being targeted in clinical trials, one of which being TIGIT. We have added this information to the results section and added the corresponding graph as supplementary figure 5.

For the putative inhibitory receptors, we believe the functional categorization can assist in selecting targets that are more likely to be successful in a therapeutic context. As we previously proposed in our perspective on functional categorization of inhibitory receptors (Rumpret et al., Nat Imm, 2020), it might be beneficial to target inhibitory receptors of different functional categories in cancer immunotherapy. Targeting a threshold receptor to lower the threshold for activation and a negative feedback receptor to lengthen and strengthen the cellular response might therefore be more effective than targeting two receptors of a single functional category. Even though we realize RNA sequencing data of in vitro stimulated immune cells is not identical to data from TILs, we have tried to characterize the functional categories expressed by TILs by extrapolating the defined functional categorization per gene from figure 2, and added the corresponding graphs as supplementary figure 4. This shows that mainly threshold receptors and some (threshold-)negative feedback receptors are expressed by the different T cell subsets, which would open the possibility of using the proposed therapeutic strategy of targeting different functional categories. However, we acknowledge that this will require further validation of expression patterns in vivo in different cancers and immune cell subsets.

**Reviewer #1 (Recommendations For The Authors):**
One comment/suggestion regarding the methodology of evaluating gene expression profiles of putative receptors: perhaps it might be important to look at clusters of genes that are co-expressed with putative inhibitory receptors.

See our reply to the suggestion above.

**Reviewer #2 (Recommendations For The Authors):**
Results section(a) "Putative ITIM/ITSM-bearing immune inhibitory receptors can be found in the human genome"i. Figure 1 could benefit from additional labeling. For example, in B, the grey line indicates 5%, etc. Additionally, in panel B&C, I assume by "predicted" the author meant using TOPCONS?ii. Figure 1B doesn't seem to be consistent with this sentence "However, for 10 out of 51, we observed ITIM/ITSM sequences in the permutated sequence up to ~25% of the time" [page 2, line 1-3], as all 51 data points in Figure 1B (under "Known" panel) are below the 0.25 horizontal line?

i. We have adjusted the figure legend to better indicate the information provided in the figures. The predicted genes are all unknown transmembrane candidates that contain an ITIM or ITSM in their intracellular domain, as determined using TOPCONS.

ii. Due to the nature of permutation testing, there is some variation in the individual likelihood values for each protein sequence. However, as they were generally below 0.25 in any given iteration, we decided to define this value as a threshold for inclusion.

(b) "AlphaFold structure predictions can assist in identifying likely functional ITIM/ITSMs"i. Readability would increase if the author indicate how pLDDT score is computed and in what range is it (between 0 and 100.)ii. Third paragraph. Can the author comment on why 80 pLDDT is chosen as the cutoff? The first sentence of this paragraph states "We found that 99 out of 101 ITIM/ITSMs of the 51 known receptors had low confidence score, i.e., less than 80 pLDDT, with an average confidence score of 49.3 pLDDT..." However, it was later stated in the Discussion, page 10, starting Line 11 "We determined a threshold of 80 pLDDT based on the average prediction scores of the ITIM/ITSMs in known inhibitory receptors....". If 99 out of 101 ITIM/ITSMs had pLDDT<80, then it seems strange that the average of the 101 is at 80pLDDT, even in the extreme where the remaining 101-99=2 ITIM/ITSMs attain the maximum pLDDT score at 100, unless the distribution of those 99 is narrowly centered around 80? A distribution of the pLDDT would help clarify.

i. The pLDDT scores are computed by AlphaFold as a way to determine how well a specific residue and/or region is expected to be modelled in three-dimensional space. We now refer to the corresponding AlphaFold publications and references therein to clarify this (10.1093/nar/gkab1061, 10.1038/s41586021-03819-2, 10.1093/bioinformatics/btt473). We also have now included the range (i.e., 0-100) in the text.

ii. The threshold of 80 pLDDT was chosen as this still encompasses all known inhibitory receptors and was not calculated based on an average of the prediction scores. In this way, we still included ITIM/ITSMs with a relatively high pLDDT, such as those observed in PD-1 and LAIR-1. The previous text ‘average prediction scores of the ITIM/ITSMs in known inhibitory receptors’ referred to the averaging of the confidence score for each of the six amino acids encompassing the ITIM/ITSM into one overall score per ITIM/ITSM. We have adjusted the text to better reflect this.

(c) "Putative inhibitory receptors are expressed across immune cell subsets"Figure S2, the last sentence in the caption (relevant for panel C) states "Cell subsets without uniquely expressed putative inhibitory receptors i.e., B cells and T cell, are excluded from the panel for clarity", but B cells and T cells are present in panel C?

Indeed, but they are only included for the cases where the cell subsets share receptor expression with other immune cell subsets. The B and T cells do not express any unique putative multi-spanning receptors, all receptors are shared with at least one other immune cell subset.

(d) "Known and putative inhibitory receptors are expressed on tumour infiltrating T cells"i. Missing panel C label in Figure 3 and S3.ii. By comparing Figure 3 and S3, it looks to me that there's not a big difference between single-spanning and multi-spanning inhibitory receptors. I wonder if the authors can comment or speculate on this similarity in addition to differences of expression among T-cell subsets. Would the similarities and differences above be explained by cancer type?

i. Figure 3 and S3 do not contain a panel C, but panel B consists of a lower (CD8+) and an upper (CD4+) subpanel, we have more clearly indicated this in the figure legend in the revised manuscript.

ii. While some T cell subsets, such as exhausted CD8+ T cells and CD4+ regulatory T cells, appear to not differ much in their expression of either single- or multi-spanning receptors, we do observe that, for example, effector memory CD4+ T cells or EMRA CD8+ T cells express single-spanning inhibitory receptors to a higher extent than multi-spanning inhibitory receptors. It is possible that these differences and similarities reflect some of the roles multi-spanning inhibitory receptors could play in regulating immune cells, for example in response to chemokines, as many chemokine receptors are multi-spanning proteins.

Data and Code availabilityAlthough the Methods section provides some context for the computational analysis and citations for relevant data, software availability and a data availability statement are lacking.

We have included a data availability statement to the data files and code in the revised manuscript.